# Kinetics of Ni, V and Fe Leaching from a Spent Catalyst in Microwave-Assisted Acid Activation Process

**DOI:** 10.3390/molecules27072078

**Published:** 2022-03-24

**Authors:** Tian Wang, Jing Ren, Annavarapu V. Ravindra, Yan Lv, Thiquynhxuan Le

**Affiliations:** 1Faculty of Metallurgical and Energy Engineering, Kunming University of Science and Technology, Kunming 650093, China; wtkust@163.com; 2Engineering Technology Research Institute of SINOPEC Catalyst Ltd., Beijing 101111, China; renj.chji@sinopec.com (J.R.); lvyan.chji@sinopec.com (Y.L.); 3Department of Physics and Nanotechnology, SRM Institute of Science and Technology, Tamilnadu 603203, India; ravindra.annavarapu@outlook.com

**Keywords:** nickel, vanadium, iron, leaching, microwave-assisted leaching, spent catalyst

## Abstract

Ni, V and Fe are the main contaminant metals that lead to the deactivation of the spent fluid catalytic cracking (SFCC) catalyst. In this work, the properties and distribution of Ni, V and Fe in the SFCC catalyst are investigated by employing EPMA-EDX, SEM and XPS techniques. The kinetics of Ni, V, Fe and Al leaching in organic and inorganic acids are studied under microwave heating. The EPMA-EDX results show that Fe and Ni mainly accumulate near the particle surface, while V eventually distributes throughout the catalyst particle. The XPS result suggests that the phase speciations of Ni in the SFCC catalyst are Ni, Ni_2_SiO_4_ and NiAl_2_O_4_, while Fe is present in a mixture of Fe_3_O_4_, Fe_2_O_3_ and Fe_2_SiO_4_. V is in the forms of V_2_O_5_ and VO_2_. Compared with oxalic acid, sulfuric acid has a better removal effect of contaminant metals, especially for Ni. The leaching kinetics results indicate that using either sulfuric acid or oxalic acid, the apparent activation energy of V is obviously lower than that of Fe and Ni, and the priority of the three contaminant metals in the removal effect is V > Fe > Ni. In addition, the leaching kinetics of contaminant metals in the microwave-assisted acid activation process are controlled by the surface chemical reaction control model.

## 1. Introduction

Fluid catalytic cracking (FCC) catalysts play a key role in the modern petroleum industry to improve the process efficiency of the conversion from heavy crude oils into lighter useful products [1,2]. However, the catalyst is deactivated over time [3]. When its activity declines to an unacceptable level, the catalyst is purged from the cracking unit and exchanged with a fresh catalyst. In such cases, the replaced catalyst is referred to as the spent FCC catalyst (SFCC), which is classified as hazardous waste. The amount of spent FCC catalyst generated is 966,000 tons per year worldwide [4] and more than 200,000 tons per year in China [5].

The deactivation of FCC catalysts is mainly due to the blockage of pore structure or the destruction of the crystalline zeolite framework by the contaminant metals (Ni, V and Fe) from crude oil during the FCC process under high temperature [3,6,7]. The landfill disposal of SFCC catalysts becomes costly and prohibitive [8] or even higher because the contaminant metals in SFCC catalysts pose a serious threat to the environment. Therefore, in order to solve the problem of SFCC catalysts, they should be transformed from “landfill disposal” into eco-friendly by “turning waste into treasure”. Among them, the regeneration and reuse of SFCC catalysts by removing contaminant metals deposited on the catalyst without affecting their catalytic performance have attracted the attention of researchers.

Hydrometallurgical processes are the most widely used to remove contaminant metals from SFCC catalysts because of their low cost, simple process, high catalytic activity [9] and environmentally friendly characteristics. Several inorganic acid solutions, such as hydrochloric acid, sulfuric acid and nitric acid, have been studied as leaching solvents to remove contaminant metals from spent catalysts [10,11,12,13]. Li et al. [14] reported that 30% V, 23.5–24.6% Fe and no Ni were extracted from the SFCC catalyst after leaching with 16% H_2_SO_4_ solution or 16% HCl solution for 1.5 h, while only 28% V, 0.8% Fe and 3.8% Ni were extracted from the SFCC catalyst after leaching with 16% HNO_3_ solution for 1.5 h. Mouna and Saroj [8] compared the effects of hydrochloric acid, sulfuric acid and nitric acid for leaching Ni and V from the SFCC catalyst and found that hydrochloric acid showed maximum leaching efficiency (39% V, 33% Ni), followed by sulfuric acid (32% V, 27.5% Ni) and nitric acid (22% V, 18% Ni), but the leaching time needed as long as 21 days in their work. However, hydrochloric acid has issues such as volatilization and difficulty utilizing waste acid, while nitric acid has a high cost. Sulfuric acid is therefore often selected as an inorganic leachate in the industry. In addition, organic acids are also a choice worth considering because they have the advantages of milder treatment conditions, avoiding secondary pollution and corrosiveness. Mouna et al. [4] found that among different organic acids (citric acid, gluconic acid, oxalic acid), oxalic acid has the highest efficiency for leaching metals (Ni and V) from a spent hydrotreating catalyst. Cho et al. [9] studied many methods (chlorination, carbonylation, water washing, oxalic washing, citric washing methods) to rejuvenate spent catalysts and indicated that the catalytic activity of the catalyst regenerated by the washing method with oxalic acid was the highest. In addition, they found that although only 10% of Ni and 26% of V were removed after washing with oxalic acid, the catalytic activity and yields toward gasoline and diesel increased.

Microwave-assisted leaching has been reported to significantly reduce the processing time and improve the leaching effect in several studies [15,16,17]. Better controllability, selective heating, rapid heating and environmentally friendly characteristics are the reasons why microwave-assisted leaching appears to be attractive for treating spent catalysts [18]. A novel microwave heating method, including a microwave-assisted acid activation step and microwave synthesis step, was proposed for the synthesis of zeolite-Y from SFCC catalysts in our work [19]. We found that the microwave-assisted acid activation process is an important step in our method, which can affect the types of zeolite products in the subsequent synthesis step. The removal of contaminant metals without destroying the zeolite-Y framework is a key objective in the microwave-assisted acid activation step. However, the characterization of contaminant metals in SFCC catalysts and their leaching behavior in inorganic and organic acids under microwave irradiation have not been studied.

In this work, the properties and distribution of contaminant metals Ni, V and Fe in SFCC catalysts have been investigated by adopting various modern analysis techniques such as SEM, EPMA-EDX and XPS studies. Two types of acids, one inorganic (sulfuric acid) and the other organic (oxalic acid) are used as leaching solutions. The kinetics of contaminant metal leaching in sulfuric acid and oxalic acid are compared under microwave heating. Combined with the characterization of the contaminant metals and their leaching kinetics, the priority of different contaminant metals in the removal effect is revealed under the action of a microwave.

## 2. Experimental

### 2.1. Experimental Work

The SFCC catalyst used in this work was collected from the industrial FCC process of a petrochemical company in Beijing, China. The raw SFCC catalyst contains Al_2_O_3_ (49.17 wt.%) and SiO_2_ (39.49 wt.%) and contaminated metals including Fe (0.44 wt.%), V (0.55 wt.%) and Ni (0.50 wt.%). The XRD results given in our previous report [19] showed that the SFCC catalyst consists of zeolite-Y as the main phase along with the active γ-Al_2_O_3_ with SiO_2_ and zeolite ZSM-5 phases. In this work, the SFCC catalyst was first calcined at 600 °C for 2 h in the air to remove coke and then cooled to room temperature.

In order to study the distribution and migration of contaminant metals, the catalyst molded in epoxy was sliced and polished to obtain EPMA results.

In each experiment, a mixture of the acid solution and 4 g of the SFCC catalyst was poured into closed polytetrafluoroethylene reaction tanks and then placed in an intelligent microwave digestion instrument (METASH, Model MWD-620, Shanghai Metash Instruments Co., Ltd., Shanghai, China) for leaching experiments. A microwave power of 400 W, a working frequency of 2450 MHz and a pressure of 2 kg/cm^2^ were set up in each experiment. The reaction solution was quickly heated to different leaching temperatures and then held at the set temperature for different leaching times under the action of a microwave. The SFCC catalysts were leached at various leaching temperatures of 50–100 °C for different leaching times of 10–60 min with a solid-to-liquid ratio of 1:10 under acid concentrations of 0.1–1.0 mol/L for both oxalic acid and sulfuric acid solutions. During the leaching process, the solution temperature is measured by non-contact penetration infrared scanning. Depending on the actual temperature of the solution, microwave power is provided intermittently to maintain the desired temperature. After leaching, the catalyst was further washed and dried at 100 °C overnight.

### 2.2. Analytical Methods

The chemical composition of the spent catalyst was determined using an inductively coupled plasma-atomic emission spectroscopy (Leeman, Prodigy7, Mason, OH, USA). The surface morphology of the spent catalyst was investigated by using the scanning electron microscopy technique (Philips XL20 ESEM-TMP, Amsterdam, The Netherlands). The depth profile of the contaminant metals in spent catalysts was investigated using an electron probe microanalyzer (EPMA: JXA8230, JEOL, Beijing, China) attached to an energy-dispersive X-ray (EDX) spectroscope. In our work, the thickness of the contamination layer and the penetration of contaminant metals, which are comparative and qualitative evaluations, are obtained by measuring the thickness of the bright area with highly contaminated metal content through an electron probe microanalyzer. The chemical states of elements were determined by using the X-ray photoelectron spectroscopy technique (PHI-5300, PHI, Waltham, MA, USA).

### 2.3. Leaching Kinetics

The reactions between contaminant metals present in the spent catalyst and acid solution are liquid-solid phase reactions. Hence, the rate of reaction is controlled by [20,21]: (a) the chemical reaction at the particle surface; (b) diffusion through the product layer; and (c) a combination of both. The dynamic equations for the surface chemical reaction control model and the diffusion control model are given as Equations (1) and (2), respectively.
1 − (1 − *x*)^1/3^ = *kt*(1)
1 − 3(1 − *x*)^2/3^ + 2(1 − *x*) = *kt*(2)

Here, *x* is the leaching rate of metals (min^−1^), *t* is the reaction time (min), and *k* is the apparent rate constant.

The apparent activation energy can be calculated using the Arrhenius equation [22].

## 3. Results and Discussions

### 3.1. Characterization of Contaminant Metals on Spent FCC Catalyst Particles

#### 3.1.1. Distribution and Migration of Contaminant Metals

The contamination layers formed by contaminant metals on the surface of SFCC catalyst particles are shown in Figure 1.

The EPMA result in Figure 1 clearly shows that contaminant metals deposit on the outer layer of the spent catalyst particles, creating rings containing a high concentration of contaminant metals around the particles. These rings with denser structures are seen brighter in Figure 1 and are called contamination layers. As shown in Figure 1b, the thickness of the contaminated layer is generally approximately 1.5 to 3 μm. When the spent catalyst is treated by the conventional acid leaching method, the contamination layer hinders the contact of the acid solvent with the contaminant metals, resulting in a low removal effect of contaminant metals. The composition of spot 1 is determined to be Ni 5.27%, Fe 3.62% and V 1.43%, while the composition of spot 3 is Ni 4.73%, Fe 0.36% and V 1.03%. The percentages “%” of the elements are by weight percentages. It can be seen from Figure 1 that the thickness of the contamination layer at spot 1 is greater than that at spot 3. This suggests that the concentration of contaminant metals on the particle surface increases with the accumulation of contaminant metals, resulting in a thicker contaminated layer.

Evidently, as expected, the bright area located at the particle surface (spot 1) has a higher content for Fe and Ni, especially Ni, compared to spot 2, which is located inside the particle. This shows that Fe and Ni mainly accumulate at and near the particle surface, which indicates that the mechanism of Fe and Ni enrichment in catalyst particles is mainly surface deposition. Unlike Fe and Ni, V eventually deposits throughout the catalyst particle, resulting in no significant difference in the content of V between the internal and external parts of the catalyst particle. This indicates that the deposition of V is distinctly different from that of Ni and Fe and occurs through the mechanism of deposition before migration. The distribution of different elements obtained by the EPMA technique is shown in Figure 2. V migrates into the catalyst particles, while Ni and Fe are mostly deposited near the exterior surface of the catalyst under the FCC reaction conditions. Compared with Fe, Ni penetrates deeper into the particles. It is estimated that Ni penetrates to approximately 12 μm, while Fe only reaches a depth of approximately 3 μm from the particle surface.

#### 3.1.2. Phase Speciation of Contaminant Metals in SFCC Catalyst

X-ray photoelectron spectroscopy was used to determine the phase speciation of contaminant metals in the SFCC catalyst, and the results are shown in Figure 3.

As shown in Figure 3a, the decomposed V *2p* peaks at 517.4 eV and 516.3 eV are characteristic of V^5+^ (V_2_O_5_) and V^4+^ (VO_2_) oxide [23], suggesting that V is present in the form of V_2_O_5_ and VO_2_ in the SFCC catalyst. The typical binding energy peaks of Fe_3_O_4_ can be found in the Fe *2p* 3/2 (710.4 eV) and Fe *2p* 1/2 (723.4 eV) spectra in Figure 3b, which indicate the presence of Fe_3_O_4_ in the SFCC catalyst [24]. Moreover, the binding energies of Fe^3+^
*2p* 3/2 in Fe_2_O_3_ and Fe^2+^
*2p* 3/2 in Fe_2_SiO_4_ are 710.9 eV and 709.0 eV, respectively, according to the literature data [25]. Therefore, the other peak positions of Fe *2p* 3/2 (710.9 eV) and Fe *2p* 1/2 (724.2 eV) in Figure 3b probably indicate a mixture of states between Fe_2_SiO_4_ and Fe_2_O_3_. The Ni *2p* 3/2 peak (Figure 3c) shows three peaks at binding energies of 852.7, 856.1 and 856.2 eV. They correspond to the characteristics of metallic Ni, Ni_2_SiO_4_ and NiAl_2_O_4_, according to the literature data [26]. In addition, the position of the XPS peak in the Al *2p* is located at 74.2 eV (Figure 3d), thereby evidencing the presence of NiAl_2_O_4_ (74.2 eV) and Al_2_O_3_ (74.3 eV) [23]. The binding energy peak of Si *2p* appeared between 102.5 and 103.5 eV in Figure 3e, which is most likely due to the presence of Fe_2_SiO_4_ (102.4 eV) and Ni_2_SiO_4_ (102.9 eV) [27].

Therefore, the XPS result suggests that V is deposited in the spent catalyst in the form of V_2_O_5_ and VO_2_, while Fe is present in a mixture of Fe_3_O_4_, Fe_2_O_3_ and Fe_2_SiO_4_. The phase speciations of Ni in the SFCC catalyst are Ni, Ni_2_SiO_4_ and NiAl_2_O_4_. It is noteworthy to mention that NiAl_2_O_4_ with a spinel structure is a very stable nickel-containing phase and will lead to a relatively low leaching rate of Ni in the leaching process. In addition, XPS results further confirm the conclusion of EPMA-EDX analysis that Si migrates from the inside to the surface of the particle and then combines with the contaminant metals Fe and Ni deposited on the surface to form Fe_2_SiO_4_ and Ni_2_SiO_4_ phases during the poisoning of the catalyst particles.

### 3.2. Leaching Behavior of Contaminant Metals in Organic and Inorganic Acids

#### 3.2.1. Removal Effects

(1) Effect of acid concentration

Figure 4 shows the effects of different acid concentrations on the removal of contaminant metals (Fe, V and Ni). The concentration ranges used for both solutions of oxalic acid and sulfuric acid were 0.1 to 1.0 mol/L in this work. Figure 5 reveals the morphology of SFCC catalysts after leaching with different acid concentrations.

Figure 4 shows that regardless of whether the organic acid or the inorganic acid is used as the leaching agent, the order of priority of the three contaminant metals in the removal effect is V > Fe > Ni. This may be because V is mainly present in the catalyst as the V_2_O_5_ phase, which is relatively easily leached by the acid solution. The phase speciations of Ni and Fe, such as NiAl_2_O_4_, Ni, Ni_2_SiO_4_ and Fe_2_SiO_4_, are difficult to remove by acid leaching. Nickel aluminate spinel NiAl_2_O_4_ is a very stable nickel compound because 90 wt.% of nickel was present in this form in the spent catalyst, and Ni in the spinel structure cannot be removed by washing [9]. As a result, although Fe and Ni mainly deposit near the external surface and V deposits throughout the catalyst particles, the removal effects of Fe and Ni are lower than that of V.

As seen in Figure 4, 32% V, 25.7% Fe and 11.6% Ni are removed from the SFCC catalyst after leaching with 0.5 mol/L sulfuric acid, and only 20.1% V, 12.6% Fe and 2.74% Ni are removed in the case of 0.5 mol/L oxalic acid. Under the same experimental conditions, inorganic acids (sulfuric acid) with stronger polar molecules are more likely to contact and react with spent catalysts under microwave irradiation than organic acids (oxalic acid), achieving better removal of contaminant metals, especially for the removal of Ni. However, the loss of Al in sulfuric acid solution is also more serious than that in oxalic acid, increasing from 4.3% (in oxalic acid) to 12.3% (in sulfuric acid). In other words, sulfuric acid is more favorable to the removal of contaminant metals than oxalic acid, but it has the disadvantage of considerable loss of Al. The specific surface area of the raw SFCC catalyst is 91.9 m^2^/g. Leaching in acid solution effectively restores the pores blocked by contaminant metals in the spent catalyst, thereby leading to an increase in the specific surface area of the spent catalyst. The specific surface areas of the samples after leaching with 0.5 mol/L sulfuric acid and with 0.5 mol/L oxalic acid are 178.5 and 113.6 m^2^/g, respectively. Compared with the specific surface area of the raw SFCC catalyst, the specific surface area of the spent catalyst after leaching with sulfuric acid and with oxalic acid increases by 94.2% and 23.6%, respectively. In addition, in each experiment, the mass of the solid products after leaching is 3.85–3.98 g, indicating that the yield (%) of the solid products after leaching is 96.2–99.5% in our work.

In addition, compared with the conventional leaching process, higher removal rates of contaminant metals, especially Ni, can be obtained at lower acid concentrations in our work. In the case of the conventional leaching process [14], only 30% V and 17% Fe were extracted from the SFCC catalyst after leaching with 16% H_2_SO_4_ for 1.5 h, or only 14.5% V was leached and no Ni was removed when using 0.5 mol/L oxalic acid.

The removal rates of contaminant metals increase with increasing concentration for both oxalic acid and sulfuric acid. However, the microstructure of the SFCC catalyst is intact for both oxalic acid and sulfuric acid with the acid concentrations of 0.1–0.5 mol/L and then begins to destroy when the acid concentration is as high as 0.6 or 1.0 mol/L (Figure 5). The flow of the catalyst particles with destroyed microstructures is susceptible to being hindered in the FCC process, which limits their reuse effect. The experimental results showed that regardless of whether the organic acid or the inorganic acid is used as the leaching agent, the preferred concentration for the acid activation process by using a microwave is 0.5 mol/L.

(2) Effect of leaching temperature and time

A concentration of 0.5 mol/L was selected as the acid concentration in these experiments. Figure 6 shows the effects of leaching temperature and time on the removal of contaminant metals (Ni, V and Fe). The ranges used for leaching time and temperature were 10–60 min and 50–100 °C, respectively, in this work. Figure 7 reveals the morphology of SFCC catalysts after leaching under different leaching times and temperatures.

It is obvious that increasing the temperature or prolonging the leaching time is beneficial to the reaction between the acid and the SFCC catalyst, thereby increasing the removal rates of contaminant metals, as shown in Figure 6. The order of priority of the three contaminant metals in the removal effect is V > Fe > Ni. Under the same experimental conditions, compared with oxalic acid, sulfuric acid has a better removal effect of contaminant metals, especially for Ni.

Figure 7a shows that the zeolite framework remains intact after leaching with the oxalic acid solution at 90 °C for 30 min. However, when sulfuric acid is used as the leaching agent (Figure 7b), although the microstructure of the overall particles is maintained, a few cracks are generated on the particle surface of the SFCC catalysts (marked by red circles). This is because oxalic acid reaches the boiling point quickly and can be more volatile than sulfuric acid when the temperature reaches 90 °C or above. As a result, the concentration of oxalic acid in the reaction solution decreases, which leads to a lower removal efficiency of contaminant metals, although the zeolite framework has not been destroyed.

#### 3.2.2. Leaching Kinetics

Two leaching kinetic models are fitted to the experimental data in Figure 6 according to Equations (1) and (2). The plots of the left-hand sides of Equations (1) and (2) against time for oxalic acid and sulfuric acid are depicted in Figure 8 and Figure 9, respectively. The kinetic parameters of different metals at various leaching temperatures using different kinetic models are listed in Table 1.

As shown in Figure 8 and Figure 9 and Table 1, a better fit is obtained using Equation (1) with a high R^2^ value (>0.94). This indicates that the control step of the reactions between acid solution and metals (Ni, V, Fe and Al) under microwave irradiation is the surface chemical reaction. To calculate the apparent activation energy (E), lnk is plotted against 1/T according to the Arrhenius equation. The result is depicted in Figure 10 with the fitted slope of (−E/R). The apparent activation energies of V, Fe, Ni and Al during the leaching process in sulfuric acid solution are calculated to be 4.17, 16.4, 32.7 and 19.2 kJ/mol, respectively. However, in the leaching in oxalic acid solution, they are 8.15, 20.7, 35.6 and 27.6 kJ/mol, respectively. These apparent activation energies can clearly explain why sulfuric acid is better than oxalic acid in removing contaminant metals under the action of a microwave.

The apparent activation energy of V in the acid leaching process is lower than that of Fe and Ni. This is why the removal effect of V is better than that of Fe and Ni concentrated on the particle surface, although V is distributed throughout the interior and surface of the particles. In addition, most of the Ni in the spent catalyst is present as a NiAl_2_O_4_ phase with a spinel structure, resulting in little effect on the removal of this contaminant metal regardless of the increased leaching time or temperature.

## 4. Conclusions

In conclusion, the properties and distribution of Fe, V and Ni in SFCC catalysts, and their leaching behavior in inorganic and organic acids under the microwave effect were successfully studied. The interesting findings are as follows:

(1) The EDAX results showed that Fe and Ni mainly accumulated at and near the particle surface, and compared with Fe, Ni penetrated deeper into the SFCC particle. V eventually distributed throughout the catalyst particle.

(2) The XPS result suggested that V deposited in the spent catalyst in the form of V_2_O_5_ and VO_2_, while Fe was present in a mixture of Fe_3_O_4_, Fe_2_O_3_ and Fe_2_SiO_4_. The phase speciations of Ni in the SFCC catalyst were Ni, Ni_2_SiO_4_ and NiAl_2_O_4_.

(3) The results of the leaching effect showed that sulfuric acid had a better removal effect of contaminant metals than oxalic acid, especially for Ni. 32% V, 25.7% Fe and 11.6% Ni were removed from the SFCC catalyst after leaching with 0.5 mol/L sulfuric acid, and only 20.1% V, 12.6% Fe and 2.74% Ni were removed in the case of 0.5 mol/L oxalic acid. Using either sulfuric acid or oxalic acid, the priority of the three contaminant metals in the removal effect was V > Fe > Ni.

(4) The leaching kinetics results indicated that the leaching reactions for contaminant metals with the acid solution under microwave irradiation were controlled by the surface chemical reaction control model.

(5) The apparent activation energies of V, Fe, Ni and Al during the leaching process in sulfuric acid solution were calculated to be 4.17, 16.4, 32.7 and 19.2 kJ/mol, respectively; and they were 8.15, 20.7, 35.6 and 27.6 kJ/mol, respectively, for the oxalic acid solution.

## Figures and Tables

**Figure 1 molecules-27-02078-f001:**
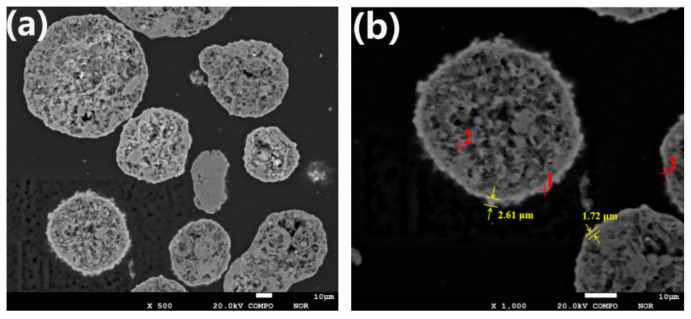
The contamination layers on the surface of SFCC catalyst particles: (**a**) cross-section of spent catalyst particles; (**b**) the thickness of the contaminated layer of spent catalyst particles.

**Figure 2 molecules-27-02078-f002:**
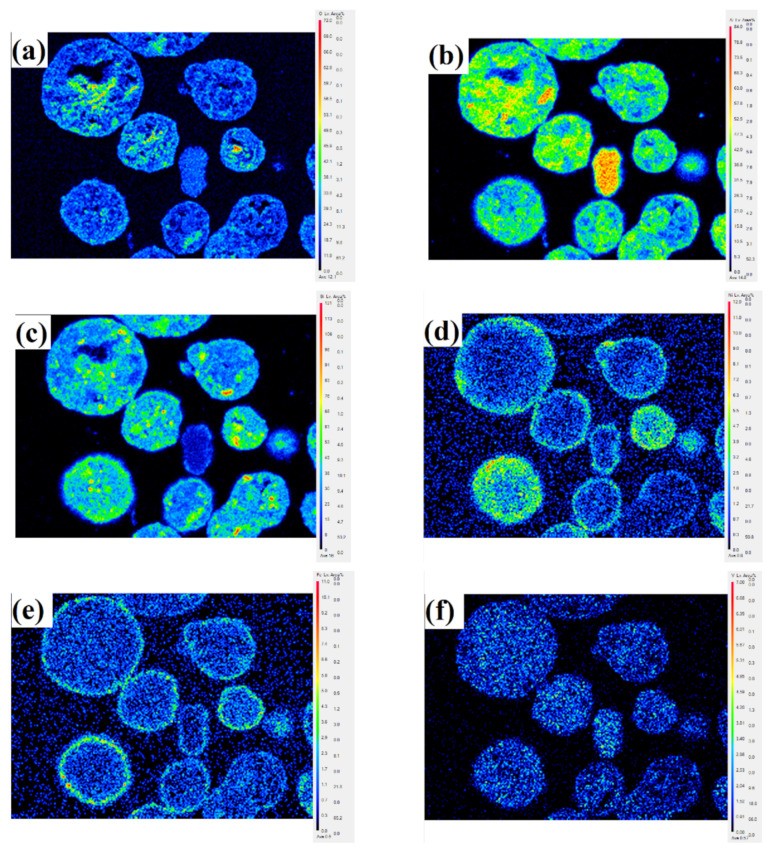
EPMA pictures showing the distribution of different elements in SFCC catalyst particles. (**a**) O, (**b**) Al, (**c**) Si, (**d**) Ni, (**e**) Fe, (**f**) V.

**Figure 3 molecules-27-02078-f003:**
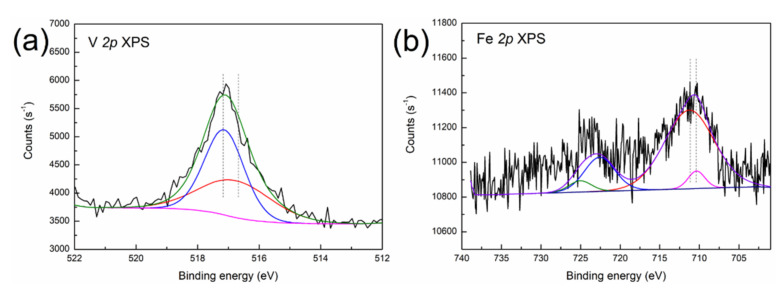
XPS spectra of (**a**) V *2p*, (**b**) Fe *2p*, (**c**) Ni *2p*, (**d**) Al *2p* and (**e**) Si *2p* in the SFCC catalyst.

**Figure 4 molecules-27-02078-f004:**
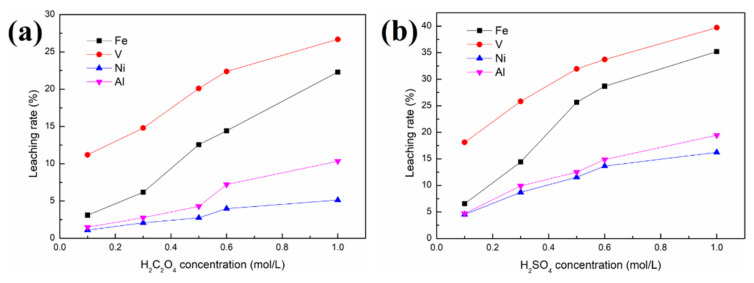
Effect of acid concentration on the removal of contaminant metals from the SFCC catalyst: (**a**) in oxalic acid; (**b**) in sulfuric acid 4 g SFCC catalyst, 80 °C, 30 min, S/L = 1:10, 400 W, 2450 MHz, 2 kg/cm^2^.

**Figure 5 molecules-27-02078-f005:**
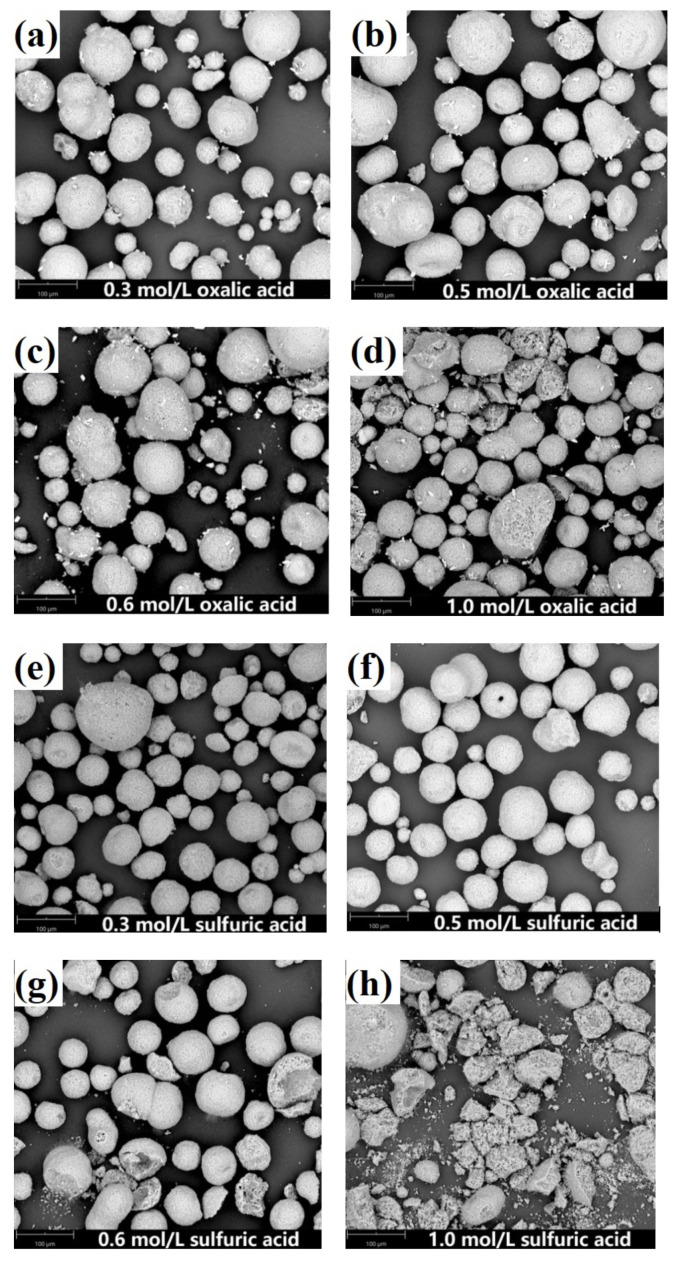
SEM images of leaching residue obtained with different acid concentrations. (**a**) 0.3 mol/L oxalic acid, (**b**) 0.5 mol/L oxalic acid, (**c**) 0.6 mol/L oxalic acid, (**d**) 1.0 mol/L oxalic acid, (**e**) 0.3 mol/L sulfuric acid, (**f**) 0.5 mol/L sulfuric acid, (**g**) 0.6 mol/L sulfuric acid, (**h**) 1.0 mol/L sulfuric acid.

**Figure 6 molecules-27-02078-f006:**
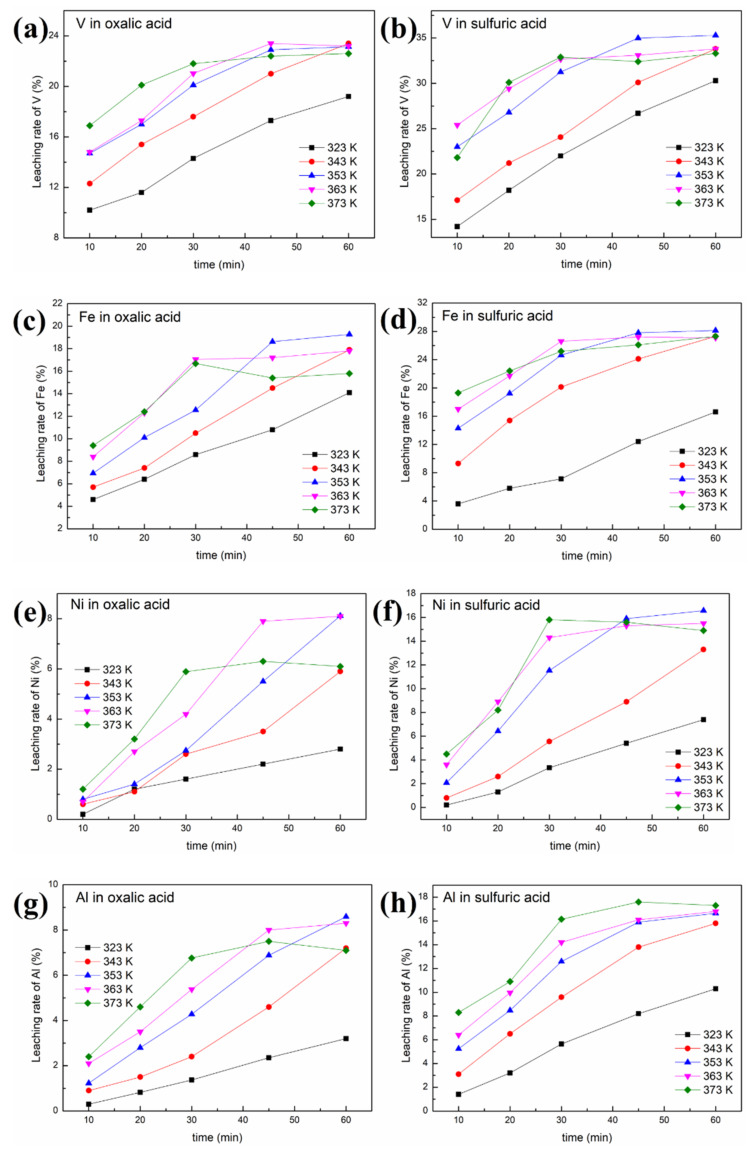
Effect of leaching temperature and time on the removal of contaminant metals from the SFCC catalyst: (**a**) leaching rate of V in oxalic acid; (**b**) leaching rate of V in sulfuric acid; (**c**) leaching rate of Fe in oxalic acid; (**d**) leaching rate of Fe in sulfuric acid; (**e**) leaching rate of Ni in oxalic acid; (**f**) leaching rate of Ni in sulfuric acid;(**g**) leaching rate of Al in oxalic acid; (**h**) leaching rate of Al in sulfuric acid (4 g SFCC catalyst, 0.5 mol/L acid concentration, S/L = 1:10, 400 W, 2450 MHz, 2 kg/cm^2^).

**Figure 7 molecules-27-02078-f007:**
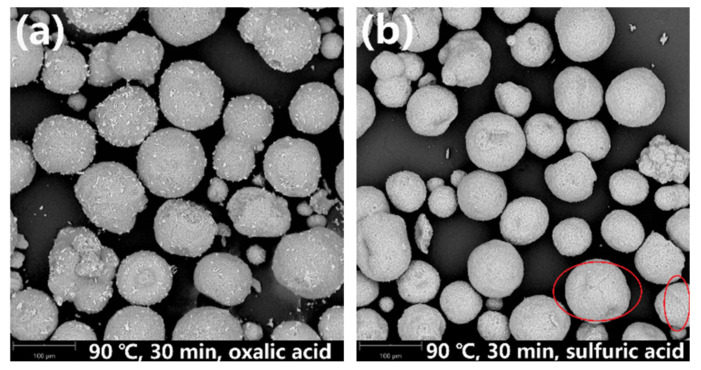
SEM images of leaching residues obtained after leaching at 90 °C for 30 min in (**a**) oxalic acid and (**b**) sulfuric acid.

**Figure 8 molecules-27-02078-f008:**
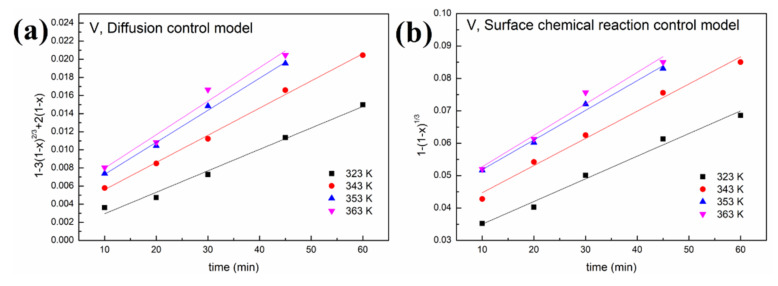
Leaching kinetics of different elements in oxalic acid: (**a**) V using the diffusion control model; (**b**) V using the surface chemical reaction control model; (**c**) Fe using the diffusion control model; (**d**) Fe using the surface chemical reaction control model; (**e**) Ni using the diffusion control model; (**f**) Ni using the surface chemical reaction control model; (**g**) Al using the diffusion control model; (**h**) Al using the surface chemical reaction control model.

**Figure 9 molecules-27-02078-f009:**
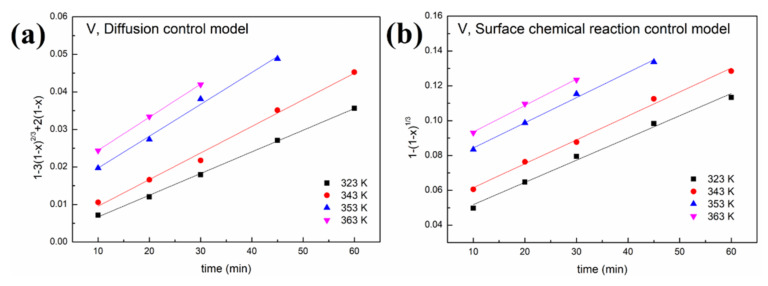
Leaching kinetics of different elements in sulfuric acid: (**a**) V using the diffusion control model; (**b**) V using the surface chemical reaction control model; (**c**) Fe using the diffusion control model; (**d**) Fe using the surface chemical reaction control model; (**e**) Ni using the diffusion control model; (**f**) Ni using the surface chemical reaction control model; (**g**) Al using the diffusion control model; (**h**) Al using the surface chemical reaction control model.

**Figure 10 molecules-27-02078-f010:**
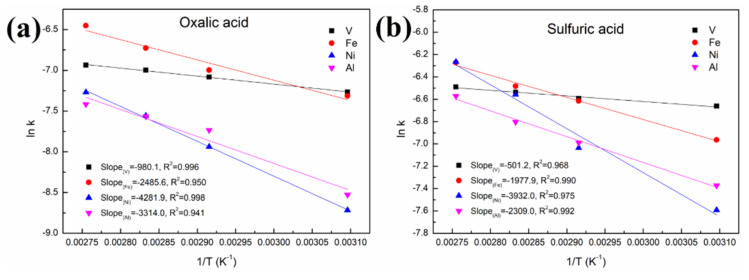
Linear fitting of (lnk) vs. (1/T) in (**a**) oxalic acid and (**b**) sulfuric acid.

**Table 1 molecules-27-02078-t001:** Kinetic parameters at various leaching temperatures using different kinetic models.

		1 − 3(1 − x)^2/3^ + 2(1 − x)			1 − (1 − x)^1/3^		
		*T* (K)	Rate Constant *k*	*R^2^*	*T* (K)	Rate Constant *k*	*R^2^*
V	oxalic acid	323	2.37 × 10^−4^	0.984	323	6.99 × 10^−4^	0.983
		343	3.00 × 10^−4^	0.995	343	8.39 × 10^−4^	0.987
		353	3.54 × 10^−4^	0.993	353	9.15 × 10^−4^	0.987
		363	3.70 × 10^−4^	0.959	363	9.72 × 10^−4^	0.961
	sulfuric acid	323	5.77 × 10^−4^	0.998	323	0.00128	0.991
		343	7.09 × 10^−4^	0.989	343	0.00137	0.993
		353	8.48 × 10^−4^	0.990	353	0.00145	0.993
		363	8.81 × 10^−4^	0.999	363	0.00152	0.995
Fe	oxalic acid	323	1.25 × 10^−4^	0.954	323	6.66 × 10^−4^	0.995
		343	2.17 × 10^−4^	0.966	343	9.15 × 10^−4^	0.993
		353	3.12 × 10^−4^	0.916	353	0.0012	0.981
		363	4.03 × 10^−4^	0.948	363	0.00158	0.991
	sulfuric acid	323	1.92 × 10^−4^	0.901	323	9.46 × 10^−4^	0.974
		343	5.096 × 10^−4^	0.993	343	0.00134	0.945
		353	6.54 × 10^−4^	0.968	353	0.00153	0.942
		363	8.22 × 10^−4^	0.985	363	0.00189	0.999
Ni	oxalic acid	323	5.18 × 10^−6^	0.985	323	1.64 × 10^−4^	0.944
		343	2.26 × 10^−5^	0.816	343	3.57 × 10^−4^	0.958
		353	4.55 × 10^−5^	0.855	353	5.22 × 10^−4^	0.969
		363	6.11 × 10^−5^	0.841	363	6.97 × 10^−4^	0.984
	sulfuric acid	323	3.86 × 10^−5^	0.924	323	5.05 × 10^−4^	0.993
		343	1.24 × 10^−4^	0.875	343	8.80 × 10^−4^	0.992
		353	2.64 × 10^−4^	0.967	353	0.00142	0.982
		363	3.43 × 10^−4^	0.932	363	0.0019	0.999
Al	oxalic acid	323	6.98 × 10^−6^	0.920	323	1.98 × 10^−4^	0.998
		343	3.46 × 10^−5^	0.823	343	4.38 × 10^−4^	0.956
		353	5.20 × 10^−5^	0.959	353	5.18 × 10^−4^	0.995
		363	6.14 × 10^−5^	0.962	363	6.01 × 10^−4^	0.978
	sulfuric acid	323	7.52 × 10^−5^	0.972	323	6.28 × 10^−4^	0.990
		343	1.82 × 10^−4^	0.985	343	9.22 × 10^−4^	0.972
		353	2.40 × 10^−4^	0.984	353	0.00111	0.979
		363	2.89 × 10^−4^	0.946	363	0.0014	0.993

## Data Availability

The data presented in this study are available on request from the corresponding author.

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
