# Peer review of "Kinetics of Ni, V and Fe Leaching from a Spent Catalyst in Microwave-Assisted Acid Activation Process"

_molecules, 2022, doi:10.3390/molecules27072078_

Round 1

Reviewer 1 Report

Dear Author,

Good work and presented well, except picture 8 and 9, all other picture should present detailed manner (Increase picture size). Scientifically, contents are good throughout the manuscript.

In Conclusion, Result mentioned point by point, it look good and easily can relate with result.

Compliments to authors. 

Author Response

Dear Professors,

The authors thank all the reviewers and editors for the valuable suggestions/comments to improve the quality of the manuscript. Below are our responses to the questions from Reviewers:

Reviewer #1: 

Good work and presented well, except picture 8 and 9, all other picture should present detailed manner (Increase picture size). Scientifically, contents are good throughout the manuscript.

In Conclusion, Result mentioned point by point, it look good and easily can relate with result.

Compliments to authors. 

As suggested by Reviewer 1, we have increased the picture size of all figures; please refer to Figs. 1~10 in the revised manuscript.

Again thank you for your time and help with the manuscript.

Kind regards!

Corresponding author:  Thiquynhxuan Le (quynhxuanlt@kust.edu.cn) 

Reviewer 2 Report

The aim of the paper is to find the optimal conditions of temperature and time for a microwave-assisted leaching procedure of an exhausted zeolite catalyst, using two different acids: sulfuric acid and oxalic acid.

The study is well structured, and conclusions are well supported by the results. I have only some suggestions for the authors in order to improve their paper, after those the paper will be ready for the publication on molecules.

  1. There is a typo at page 2 line 90 “ssolid”
  2. The leaching procedure is not very clear. I did not understand if the samples are heated to the desired temperature in an oven and then put in the microwave or if it is the process in the microwave that brings the samples to that temperature. In the second case, since multiple temperatures have been used, are the 400 W of power not fixed but intermittent to maintain the desired temperature? You should clarify this part in the methods section.
  3. Page 5 line 112 “min-1” the -1 should be superscripted
  4. The captions of all the figures should be improved to be more exhaustive, specifying what we are observing and distinguishing between image a, image b, etc.
  5. You talk about the thickness of the contamination layer. But the layer is very small if compared to the scale of the image and quite blurry. Have you use a quantitative method? For example, looking at the grey values of the profile, o another method? if not, you should specify that it is a comparative and qualitative evaluation.
  6. Page 5 line 128, the percentages “%” of the elements are by weight or atomic percentages?
  7. How do you quantify the penetration of the metals by the EPMA maps?
  8. In figure 2, EPMA maps, you should insert a color scale.
  9. In figure 4 and figure 6 you report the leaching rate, that should have a unit of 1/min, instead the unit is %. Is it the total removal percentage? The removal per minute (%/min)?

Reviewer 3 Report

Manuscript ID: molecules-1625851

Title: Kinetics of Ni, V and Fe leaching from a spent catalyst in the microwave-assisted acid activation process

Authors: Tian Wang et al.

Line 25. This information about what year?

Line 48-52. Authors must describe these three acids and add information about metal extraction degree, temperatures, leaching time, S/L ratio, etc. I ask the authors to compare these methods in terms of the efficiency of catalyst regeneration. Add more new references in this area.

Line 53-58. What organic acids have been compared with? Add numerical comparison values.

Section 2.3. Authors must add kinetic equation for external diffusion: X = kt.

Line 121-133. I agree with the authors that Fe and Ni are on the surface of the particles, however, V is inside the particles. This is clearly seen in Figure 2.

Figure 3. Authors must write information about (a) – (e) figures.

Figure 4. Enlarge figures and improve it resolution up to 300 DPI

Figure 6. The data in figures 6 are not reliable at temperatures of 363-373 K. The authors write that the acid boiled away, however, in order to make reliable studies, it is necessary to use an off-gas cooler. In this way, acid losses can be reduced. Authors must provide a layout of the experimental setup and improve experimental data.

Figure 8-9. Kinetic curves should exit from the "0" point. Only experimental data are used in these graphs, which is an error.

Figure 8-9.  The authors must show a comparison of the regression coefficients (R2) for each equation and based on these data, select data for calculating the activation energy. This information is not included in the article.

This is a hydrometallurgical study. Authors must provide the following information:

  • chemical composition, XRD, LD, BET (specific surface area - m2/g) of raw material (SFCC catalyst) and samples after leaching with both acids.
  • The extraction of vanadium, iron, and nickel is very low. How can we talk about the effectiveness of this process? To what extent can these catalysts (after leaching) be reused?
  • How can the process be intensified?
  • Why didn't the authors show particle SEM mapping after leaching?
  • Authors must show data of the yield (%) of the solid products after leaching.

The conclusions do not contain data on the extraction of vanadium, iron, and nickel.

The list of references is small and contains old links. Authors need to add 5-7 links from 2015 to 2022.

In this form, the article does not correspond to the  Molecules scientific level and should be rejected.

Round 2

Reviewer 3 Report

The authors answered some questions in detail. However, some remain without answered:

1) Authors didn't add information about solid residue after leaching process (by two acids).
2) Authors didn't show particles by SEM mapping after leaching. This information is necessary to understand the mechanism of metal extraction during the leaching process.

But the main problem are the data in Figures 6 and Figures 8:

1) The authors say that during microwave heating, some acid evaporates. That's quite possible. However, then the acid concentration is not equal to 0.5 M (see line 267), which means that it is impossible to compare the values ​​at 353-373 K and 323-343 K. Since in addition to the difference in temperature, there is a difference in the acid concentration. Moreover, if part of the solution evaporated and this affected the extraction, then the S:L ratio also changed.

In this case, we can see completely different conditions that cannot be compared on the same figure. In this leaching process, the dependence is linear, which means that with increasing temperature, the extraction of metals should increase. The low extraction at high temperatures is associate with a significant overheating of the solution, which means a decrease of the acid concentration and a decrease in the S:L ratio. The Figure 6 is not reliable.

2) The authors provided links to several articles in high impact journals to confirm their methodology for calculating the kinetics. However, unfortunately, in these journals it is also calculated incorrectly. This is a common mistake that occurs in many articles. The fitting line must leave point "0". Read more here: O. Levenspiel Chemical Reaction Engineering (3rd edition) https://www.academia.edu/23659798/Chemical_Reaction_Engineering_3rd_Edition_by_Octave_Levenspiel